# G-Protein Coupled Receptors in Human Sperm: An In Silico Approach to Identify Potential Modulatory Targets

**DOI:** 10.3390/molecules27196503

**Published:** 2022-10-01

**Authors:** Pedro O. Corda, Joana Santiago, Margarida Fardilha

**Affiliations:** Department of Medical Sciences, Institute of Biomedicine-iBiMED, University of Aveiro, 3810-193 Aveiro, Portugal

**Keywords:** G-protein coupled receptors, spermatozoa, male fertility, bioinformatics

## Abstract

G protein-coupled receptors (GPCRs) are involved in several physiological processes, and they represent the largest family of drug targets to date. However, the presence and function of these receptors are poorly described in human spermatozoa. Here, we aimed to identify and characterize the GPCRs present in human spermatozoa and perform an in silico analysis to understand their potential role in sperm functions. The human sperm proteome, including proteomic studies in which the criteria used for protein identification was set as <5% FDR and a minimum of 2 peptides match per protein, was crossed with the list of GPCRs retrieved from GLASS and GPCRdb databases. A total of 71 GPCRs were identified in human spermatozoa, of which 7 had selective expression in male tissues (epididymis, seminal vesicles, and testis), and 9 were associated with male infertility defects in mice. Additionally, ADRA2A, AGTR1, AGTR2, FZD3, and GLP1R were already associated with sperm-specific functions such as sperm capacitation, acrosome reaction, and motility, representing potential targets to modulate and improve sperm function. Finally, the protein-protein interaction network for the human sperm GPCRs revealed that 24 GPCRs interact with 49 proteins involved in crucial processes for sperm formation, maturation, and fertilization. This approach allowed the identification of 8 relevant GPCRs (ADGRE5, ADGRL2, GLP1R, AGTR2, CELSR2, FZD3, CELSR3, and GABBR1) present in human spermatozoa that can be the subject of further investigation to be used even as potential modulatory targets to treat male infertility or to develop new non-hormonal male contraceptives.

## 1. Introduction

Spermatozoa are highly differentiated haploid cells generated during spermatogenesis, a complex network of mitotic and meiotic divisions that occurs within seminiferous tubules in the testis [1]. Despite being morphologically differentiated when they leave the seminiferous tubules, spermatozoa are functionally immature and unable to fertilize oocytes, requiring two additional maturation steps: epididymal maturation and capacitation [2,3,4]. The first occurs in the epididymis, where spermatozoa acquire progressive motility and their morphological features are tuned [2]. The epididymal biochemical environment induces changes in intracellular concentrations of Ca^2+^, cAMP, and pH, promoting a shift in protein phosphorylation status, which is fundamental for motility acquisition [2,5,6]. The second maturation round, capacitation, occurs in the female reproductive tract, making spermatozoa capable of attaching to and penetrating the oocyte zona pellucida [3,4]. During this process, spermatozoa acquire hyperactivated motility, membranes become more fluid, and intracellular levels of Ca^2+^, cAMP, and pH change leading to the activation of protein kinases [7,8,9]. The successful completion of these processes allows spermatozoa to reach and fertilize the oocyte, the main goal of the sperm life journey.

G protein-coupled receptors (GPCRs), the largest family of receptors described, are seven-transmembrane domain proteins that are coupled to different G protein subtypes or arrestins [10,11,12]. These receptors detect molecules, such as hormones, neurotransmitters, chemokines, or odorants [13]. The binding of these ligands induces conformational changes in the transmembrane and intracellular domains of the receptor, allowing its interaction with heterotrimeric G proteins or with arrestins [14,15]. After activation, GPCRs act as guanine nucleotide exchange factors (GEFs) for the α subunits of heterotrimeric G proteins, catalyzing the release of GDP and the binding of GTP for G-protein activation. Activation of different G proteins (e.g., Gi/o, Gs, G12/13, or Gq/11 subtypes) affects several cellular processes through different downstream effectors, such as adenylyl cyclase (AC), RhoGEF, or phospholipase C (PLC) [12,16,17,18,19]. Active GPCRs may also be phosphorylated by GPCR kinases (GRKs), a family of protein kinases that phosphorylate specific serine/threonine residues of GPCRs. GPCR phosphorylation leads to arrestin recruitment and activation [20,21,22,23], leading to receptor desensitization and also mediating a second wave of signaling independent of G proteins [22,23,24,25,26,27]. 

It is known that GPCRs regulate a variety of physiological and pathological mechanisms, and their role in cardiovascular and neuronal disease, diabetes, and other disorders has been investigated [24,28,29,30]. Furthermore, GPCRs represent an important and attractive group of drug targets, with more than 470 GPCR-targeted drugs currently in the clinical market [31,32,33]. Being key players in cell signaling events, the GPCRs’ presence and function in human spermatozoa are poorly addressed. Indeed, only a couple of articles are focused on this topic [34,35,36,37,38,39,40], and, to the best of our knowledge, no systematic and comprehensive review on this topic exists in the literature. The main objective of the present work was to identify the GPCRs present in human spermatozoa and to perform an in silico analysis to understand their potential role in sperm-related functions. Through this approach, it was possible the identification of relevant sperm GPCRs that can be the subject of further investigation (i) to be used as potential modulatory targets to treat male fertility or (ii) to develop new non-hormonal male contraceptives. 

## 2. Results and Discussion

### 2.1. GPCRs in Human Sperm

A total of 825 GPCRs were retrieved from GLASS and GPCRdb databases (Appendix A). This list was cross-compared with the human sperm proteome previously published [41], revealing the existence of 71 GPCRs in human spermatozoa (Figure 1a; Table 1). From those, only type-1 angiotensin II receptor (AGTR1) [42], AGTR2 [43,44], glucagon-like peptide 1 receptor (GLP1R) [45], and olfactory receptor 4C13 (OR4C13) [40] were previously studied in human spermatozoa. According to the GPCRdb [46], the human sperm GPCRs were members of five classes: class A, rhodopsin (n = 42); class B1, secretin (n = 2); class B2, adhesion (n = 16), class C, glutamate (n = 10); and class F, frizzled (n = 1) (Figure 1b; Table 1). Considering that the human sperm proteome used [41] only included proteomic studies performed using ejaculated human spermatozoa. in which a false discovery rate (FDR) <5% of protein identification was set, and proteins identified with at least two peptides were included, some GPCRs identified by other techniques were excluded. For instance, some GPCRs previously identified through immunocytochemistry and Western blot techniques in human spermatozoa [34,35,37,38,39,40] were not identified through our analysis. To maintain the proposed methodology, these GPCRs were not considered in the subsequent analyses. 

The selective tissue expression and male reproductive phenotypes associated with the 71 GPCRs identified were searched in the Human Protein Atlas (HPA) and Mouse Genome Informatics (MGI) databases, respectively. Among those, 7 had selective expression in male tissues, and 9 were associated with male infertility defects in mice (Table 2). ADGRG2 is enriched in the epididymis, CYSLTR2, EDNRA, FZD3, and NPFFR2 are enhanced in the seminal vesicle, CELSR1 is enriched in the seminal vesicle, and GRM2 is enhanced in the testis (Table 2). Regarding male reproductive phenotypes: ADGRV1, ADRA2A, ADGRG1, CELSR1, and VIPR2 were associated with abnormal testis morphology; F2R, GRM1 were associated with abnormal fertility/fecundity; ADGRG2 was associated with absent vas deferens and azoospermia; GRM1, ADGRG1, and HTR2B were associated with reduced fertility; and GRM1 and VIPR2 were associated with male infertility (Table 2). Despite not being described in the MGI database, prosaposin receptor GPR37 null mice presented abnormal testis morphology, altered spermatogenesis, and decreased sperm concentration and motility [47]. Additionally, Agtr1 knock-out mice had abnormal sperm morphology and male infertility, probably due to abnormal spermatogenesis [48]. 

ADGRG2 has been pointed out as a potential target for the development of a non-testicular, non-steroid male contraceptive [49]. ADGRG2 is an adhesion G-protein coupled receptor highly expressed in efferent ducts and proximal epididymis, being a key player in the signaling pathways that promote fluid reabsorption in the efferent ductus and sperm maturation [50,51,52]. In men, X-linked ADGRG2 deleterious mutations have been associated with congenital bilateral absence of the vas deferens [53], one of the causes underlying obstructive azoospermia. Those men presented normal spermatogenesis and were able to conceive a child through assisted reproduction techniques [53]. Nevertheless, the ADGRG2 role in sperm physiology is unknown. A proteomic study revealed increased ADGRG2 levels in spermatozoa from asthenozoospermic men [54], which may suggest a role of this receptor in sperm motility. On the other hand, ADGRG1, ADGRV1, ADRA2A, and VIPR2 were associated with histological testicular defects and impaired spermatogenesis, which make them unattractive targets for the development of drugs for male contraception since they can negatively and permanently impact the testicular environment and thus, spermatogenesis and sperm maturation. HTR2B, GRM1, and F2R were already associated with reduced or abnormal fertility; however, how they lead to this phenotype is still unknown. Hence, more studies focusing on the role and function of these GPCRs in human sperm physiology and fertilization should be performed.

### 2.2. Biological Processes Associated with Sperm GPCRs

An enrichment analysis was performed to identify significant biological processes associated with the 71 sperm GPCRs. Ninety-one gene ontology (GO) terms related to biological processes were retrieved (Appendix A). As expected, the most significant terms were “G protein-coupled receptor signaling pathway” (GO:0007186, *p*-value = 0.0) and “G protein-coupled receptor activity” (GO:0004930, *p*-value = 0.0). Since spermatozoa are a highly specific cell type, the GO term list was filtered to highlight relevant processes for sperm physiology. A total of 30 GO terms were retrieved (Appendix A), being grouped into 13 generic classes: adenylate cyclase modulating GPCR (GO:0007188; GO:0007193; GO:0007189), glutamate receptor (GO:0007196; GO:0001640; GO:0007215), adrenergic receptor (GO:0071880; GO:0071875), GPCR coupled to cyclic nucleotide second messenger (GO:0007187), cyclic nucleotide signaling (GO:0019935), phospholipase C activity (GO:0007200; GO:0051482), regulation of phospholipase activity (GO:0010518; GO:1900274; GO:0010863; GO:0010517), regulation of cyclase activity (GO:0045761; GO:0007194; GO:0031280; GO:0031279; GO:0007190), phosphatidylinositol 3-kinase activity (GO:0035004; GO:0043551), PKA signaling (GO:0010737), PKC signaling (GO:0070528; GO:0090037; GO:0090036), sodium homeostasis (GO:0055078), and Wnt signaling (GO:0060071; GO:0035567) (Figure 1c; Appendix A). Forty sperm GPCRs were associated with those categories (Figure 1c), which may indicate that those GPCRs have a role in the retrieved biological processes. 

Although this analysis did not reveal any association between GPCRs and sperm-specific processes, ADRA2A, AGTR1, AGTR2, FZD3, and GLP1R have been implicated in sperm-specific functions. The ADRA2A stimulates the cAMP production and accelerates mouse sperm capacitation [55]. AGTR1 also stimulates mammalian sperm capacitation, acrosome reaction, and motility [42,56,57,58,59,60]. In men, increased AGTR2 expression levels were positively correlated with progressive motility, suggesting that sperm motility may be regulated by this GPCR [43,44]. In porcine spermatozoa, FZD3 was reported to have a regulatory role in in vitro capacitation and acrosome exocytosis [61]. GLP1R stimulation by exenatide (a GLP1R agonist) increased human sperm motility and glucose metabolism [45]. Additionally, high fat-induced obese mice treated with exenatide presented improved sperm quality regarding motility, mitochondrial metabolism, and DNA integrity parameters [62]. Nevertheless, since exenatide treatment may directly impact sperm quality, the mice’s weight reduction may also contribute to sperm quality improvement. Considering their roles in sperm-specific functions, AGTR1, AGTR2, FZD3, and GLP1R are potential targets for further investigation and to modulate sperm function. 

### 2.3. Sperm GPCRs Interactions

To investigate whether GPCRs present in human sperm interact with relevant proteins for sperm physiology, a protein-protein interaction (PPI) network for the human sperm GPCRs was retrieved using the IMEx data (through the Cytoscape). Only 47 sperm GPCRs had described interactions and were included in this network, for which 212 sperm interactors were retrieved, with a total of 270 interactions among sperm GPCRs and those interactors (Figure 2, Appendix A). Regarding the node degree (i.e., number of connected edges to a node in the network), the top 10 GPCRs with a higher degree were highlighted: VIPR2, AGTR1, N-formyl peptide receptor 2 (FPR2), P2Y purinoceptor 12 (P2RY12), GPR37, CELSR2, CYSLTR2, alpha-1D adrenergic receptor (ADRA1D), HTR2B, and gamma-aminobutyric acid type B receptor subunit 1 (GABBR1) (bigger green nodes in Figure 2). This result highlighted that the same GPCRs may be involved in different signaling pathways. A good example of that is the AGTR1, which was associated with capacitation, acrosome reaction, and motility in mammalian sperm, all distinct sperm-related processes [42,56,57,58,59,60]. 

Next, to infer the potential role of the GPCRs in sperm physiology, the interactors were searched in the UniProt database to identify relevant biological processes to sperm physiology. Of the 212 identified GPCR interactors, 49 were associated with the following processes: spermatogenesis, cilium organization, cytoskeleton organization, Ca^2+^ ion concentration, the release of sequestered Ca^2+^ ion, voltage-gated Ca^2+^ channels, ion homeostasis, ion transport, energy metabolism, motility, sperm/oocyte interactions, acrosome reactions, and fertilization (Figure 3, Appendix A). The identified processes are associated with sperm formation and maturation, as well as with fertilization. Only 24 sperm GPCRs were associated with these 49 interactors (Figure 3). ADGRC2, ADGRG1, ADGRE5, ADGRV1, AGTR1, gastrin/cholecystokinin type B receptor (CCKBR), CELSR1, CELSR2, CELSR3, GABBR1, P2RY12, taste receptor type 2 member 7 (TAS2R7), and VIPR2 interact with proteins related to sperm-specific processes (motility, sperm/oocyte interactions, acrosome reaction, and fertilization) (Figure 3). 

The use of small molecules that target GPCRs involved in sperm motility regulation (for example, P2RY12 and TAS2R7) may be a useful strategy for developing non-hormonal male contraceptives targeting only post-testicular sperm maturation, stopping sperm. Additionally, the use of natural or synthetic drugs that activate or inhibit GPCRs associated with low sperm motility has the potential to be used as an additive in sperm cryopreservation and preparation media to improve the motility of samples used for artificial insemination and in vitro fertilization. On the other hand, the modulation of ADGRV1, AGTR1, CCKBR, and VIPR2, which interact with proteins related to spermatogenesis, is considerably more difficult since (i) the drugs used need to cross the blood-testis-barrier to reach the germ cells niche; (ii) the spermatogenic process may be irreversibly impacted; (iii) other signaling pathways may be activated or inhibited, resulting in unexpected phenotypes. Both Figure 2 and Figure 3 showed that some interactors are common to different GPCRs, which suggests two hypotheses: (i) different GPCRs can modulate the same biological process in the sperm (for example, CELSR1, ADGRV1, and CELSR2); (ii) the common interactors may be involved in different signaling pathways that are dependent on the activated GPCRs (for example, ABL1 and PRKAG1). Considering this, caution should be taken when selecting the target to modulate to avoid disturbing other important signaling pathways. Therefore, the selection of the best GPCR to modulate, both for infertility treatment or for contraception, should take into consideration its involvement in other physiological processes in sperm and even in other cell types. The selection of a sperm-specific GPCR can overcome this limitation; however, at least in this study, none of the GPCRs identified are exclusively present in sperm. 

## 3. Materials and Methods

### 3.1. Collection of GPCRs in Human Spermatozoa

To collect the human GPCRs, the GPCR-Ligand Association (GLASS; Ann Arbor; USA) and GPCRdb (Copenhagen, Denmark) databases were used (downloaded on 7 March 2022) [46,63]. To avoid redundancy, all GPCRs were mapped in the UniProt database (Cambridge, UK) (downloaded on 7 March 2022), resulting in the final list of human GPCRs (Appendix A). Through the JVenn tool [64], a Venn diagram analysis was performed to identify GPCRs in human sperm (Table 1). The human sperm proteome was obtained by Santiago et al. [41]. 

### 3.2. Detailed Molecular Insight of the Sperm GPCRs

To identify sperm GPCRs with selective expression in the male reproductive tract, the Human Protein Atlas (HPA; Stockholm, Sweden) (version 21.0, downloaded on 10 March 2022) was used [65]. Information related to male infertility defects and phenotypes was retrieved from the Mouse Genome Informatics (MGI) database (downloaded on 10 March 2022) [66]. To gain molecular insight into the sperm GPCR functions, a gene ontology (GO) enrichment analysis was performed for biological processes (Appendix A). This analysis was performed through the ClueGO plug-in (version 2.5.8; Bethesda, MD, USA) in Cytoscape (version 3.8.2; Bethesda, MD, USA) [67]. The GO range was fixed between 3 and 8, and the perfusion option was used. Only terms with a *p*-value < 0.05 were considered. Lastly, all sperm GPCRs were searched in the PubMed database (Bethesda, MD, USA) to identify studies that report their role in mammalian spermatozoa (search performed before 24 May 2022). 

### 3.3. Identification of Sperm GPCRs Interactome

To predict the potential roles of the sperm GPCRs, a protein-protein interaction (PPI) network was imported from the IMEx database through Cytoscape (version 3.8.2; Bethesda, MD, USA) [67,68] (downloaded on 10 March 2022). The imported PPI network was filtered by taxonomic name (“human”), and only sperm proteins [41] were kept. Duplicate edges and self-loops were removed. The topological properties of the network were assessed using the Network Analyzer tool [69]. To gain molecular insight into the sperm GPCR interactors (Appendix A), the GO terms related to biological processes were collected from the UniProt database (Cambridge, UK) (downloaded on 12 March 2022).

## 4. Conclusions and Future Perspectives

The identification of GPCRs and their ligands/interactors in human sperm, and the characterization of their functions and downstream signaling pathways in sperm physiology, may provide future directions for new developments in the treatment of male infertility. To date, some GPCRs have been identified in mammalian sperm and implicated in sperm-related processes such as motility and acrosome reactions. In this work, we applied a bioinformatic approach to identify human sperm GPCRs and predict their function. To our best knowledge, this study was the first approach to list which GPCRs are presented in human spermatozoa. As stated above, some sperm GPCRs identified in individual studies were not included in the sperm proteome. This is justified by the evolution of the proteomic field, where the early studies only allowed the identification of a small number of proteins due to restrictive criteria [70]. Further sperm proteomic analyses may provide new updates in sperm proteome and consequently lead to the identification of more GPCRs in human spermatozoa. Additionally, proteomics can be useful for identifying sperm-specific GPCRs, which would be good targets to modulate sperm function(s). 

Taking together all the information collected from the searched databases, we believe that at least 8 sperm GPCRs—ADGRE5, ADGRL2, GLP1R, AGTR2, CELSR2, FZD3, CELSR3, and GABBR1—are strong candidates for being explored in further research, not only to characterize their role in sperm but also to be considered potential targets for drug therapies. Those 8 GPCRs were directly associated with sperm functions or interacted with proteins with known functions in those processes. Milardi and colleagues hypothesized that OR4C13 may be involved in acrosome reaction and motility [40], which also makes it a potential research target. Nonetheless, our results did not associate OR4C13 with the mentioned sperm functions. ADGRG1, ADGRV1, CELSR1, and VIPR2 were implicated in sperm-related processes (Figure 3), but they were also associated with spermatogenesis (Figure 3) and testicular histological alterations (Table 2). Despite those GPCRs being potential targets for modulation in sperm, their function should be carefully evaluated in the context of the testicular environment. 

One limitation of the present analysis is the near absence of sperm-specific information in databases. Most of the identified GPCRs (and even the identified interactors) are exhaustively explored in somatic cells, but their roles are poorly described in spermatozoon. Consequently, it is possible to draw conjectures about the potential role of the identified GPCRs, but additional experimental studies are needed for a complete characterization of their functions, interactions, and signaling pathways in human sperm.

To summarize, despite the activation/inhibition of specific GPCRs associated with sperm motility, acrosome reaction, or fertilization being particularly interesting for male contraception, more studies should be conducted to clarify the role of these GPCRs in human spermatozoa and the implications of their modulation for the whole organism. Additionally, we should take advantage of the several drugs targeting GPCRs already approved and on the market. After the identification of relevant GPCRs involved in poor sperm quality and infertility, high-throughput drug screening should be performed to assess the impact of these compounds in spermatozoa as a first step. The development of more potent and specific drugs targeting sperm GPCRs should then be investigated. 

## Figures and Tables

**Figure 1 molecules-27-06503-f001:**
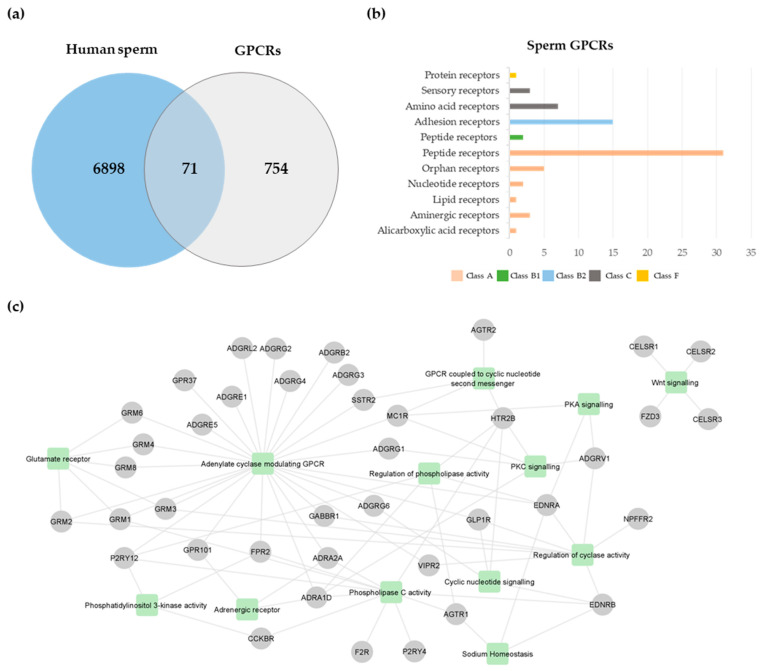
Sperm GPCRs. (**a**) Venn diagram illustrates the cross-comparison between the human sperm proteome and human GPCRs. For the subsequent analysis, only GPCRs identified in sperm were considered. (**b**) Classification of sperm GPCRs. (**c**) Integrative network of relevant processes associated with sperm GPCRs. Each node represents a GPCR (represented by gene name), and each square represents a class of biological processes.

**Figure 2 molecules-27-06503-f002:**
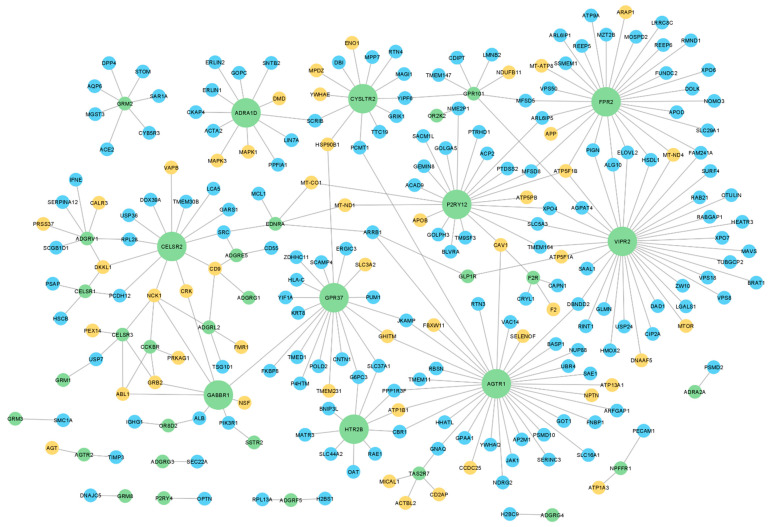
Protein-protein interaction network of sperm GPCRs. The green nodes represent sperm GPCRs (n = 47), the blue nodes represent their interactors (n = 163), and the orange nodes represent the interactors associated with relevant biological processes to sperm physiology (n = 49). The bigger green nodes correspond to the top 10 sperm GPCRs with a higher degree. All proteins are represented by the gene name.

**Figure 3 molecules-27-06503-f003:**
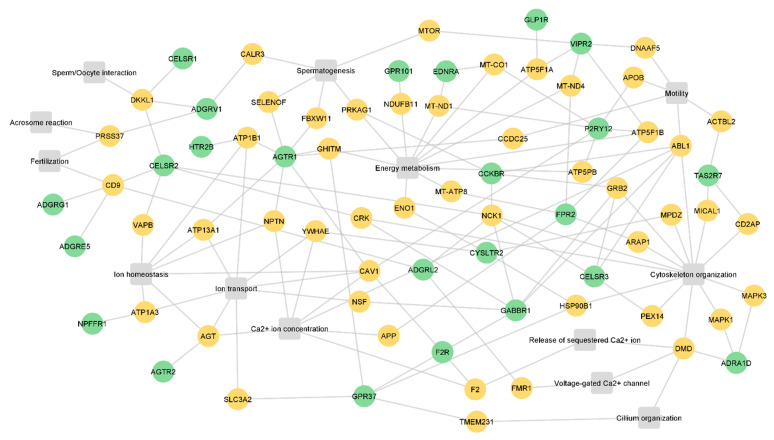
Integrative network of the relevant interactors and sperm GPCRs. The green nodes represent sperm GPCRs (n = 24), the orange nodes are the interactors associated with relevant biological processes to sperm physiology (n = 49), and the grey squares are the biological processes. All proteins are represented by the gene name.

**Table 1 molecules-27-06503-t001:** G-protein coupled receptors (GPCRs) identified in human spermatozoa. The sperm GPCRs were divided into families, and the subfamily for each receptor is indicated according to the GPCRdb.

UniProtKB	Gene Name	Protein Names	Subfamily
**Class A: Rhodopsin (n = 42)**			
Q9BXC0	HCAR1	Hydroxycarboxylic acid receptor 1	Alicarboxylic acid receptors
P41595	HTR2B	5-hydroxytryptamine receptor 2B	Aminergic receptors
P25100	ADRA1D	Alpha-1D adrenergic receptor	Aminergic receptors
Q9NS75	CYSLTR2	Cysteinyl leukotriene receptor 2	Lipid receptors
Q9H244	P2RY12	P2Y purinoceptor 12	Nucleotide receptors
P51582	P2RY4	P2Y purinoceptor 4	Nucleotide receptors
Q9NQS5	GPR84	G-protein coupled receptor 84	Orphan receptors
Q96P66	GPR101	Probable G-protein coupled receptor 101	Orphan receptors
Q5T848	GPR158	Probable G-protein coupled receptor 158	Orphan receptors
O15354	GPR37	Prosaposin receptor GPR37	Orphan receptors
Q9P1P5	TAAR2	Trace amine-associated receptor 2	Orphan receptors
P24530	EDNRB	Endothelin receptor type B	Peptide receptors
P25101	EDNRA	Endothelin-1 receptor	Peptide receptors
P32239	CCKBR	Gastrin/cholecystokinin type B receptor	Peptide receptors
Q01726	MC1R	Melanocyte-stimulating hormone receptor	Peptide receptors
Q9GZQ6	NPFFR1	Neuropeptide FF receptor 1	Peptide receptors
Q9Y5X5	NPFFR2	Neuropeptide FF receptor 2	Peptide receptors
P25090	FPR2	N-formyl peptide receptor 2	Peptide receptors
P25116	F2R	Proteinase-activated receptor 1	Peptide receptors
P30874	SSTR2	Somatostatin receptor type 2	Peptide receptors
P30556	AGTR1	Type-1 angiotensin II receptor	Peptide receptors
P50052	AGTR2	Type-2 angiotensin II receptor	Peptide receptors
Q9GZK3	OR2B2	Olfactory receptor 2B2	Peptide receptors
Q8NGT1	OR2K2	Olfactory receptor 2K2	Peptide receptors
Q8NH16	OR2L2	Olfactory receptor 2L2	Peptide receptors
Q8NG97	OR2Z1	Olfactory receptor 2Z1	Peptide receptors
Q8NGL6	OR4A15	Olfactory receptor 4A15	Peptide receptors
Q8NGP0	OR4C13	Olfactory receptor 4C13	Peptide receptors
Q8NH41	OR4K15	Olfactory receptor 4K15	Peptide receptors
Q8NGK0	OR51G2	Olfactory receptor 51G2	Peptide receptors
Q8NGH9	OR52E4	Olfactory receptor 52E4	Peptide receptors
Q8NGH8	OR56A4	Olfactory receptor 56A4	Peptide receptors
Q8NHB8	OR5K2	Olfactory receptor 5K2	Peptide receptors
A6NET4	OR5K3	Olfactory receptor 5K3	Peptide receptors
A6NMS3	OR5K4	Olfactory receptor 5K4	Peptide receptors
Q8NGP4	OR5M3	Olfactory receptor 5M3	Peptide receptors
Q8NGP3	OR5M9	Olfactory receptor 5M9	Peptide receptors
Q8NGE1	OR6C4	Olfactory receptor 6C4	Peptide receptors
Q8NG99	OR7G2	Olfactory receptor 7G2	Peptide receptors
Q9GZM6	OR8D2	Olfactory receptor 8D2	Peptide receptors
Q8NG78	OR8G5	Olfactory receptor 8G5	Peptide receptors
Q8NGP2	OR8J1	Olfactory receptor 8J1	Peptide receptors
**Class B1: Secretin (n = 2)**			
P43220	GLP1R	Glucagon-like peptide 1 receptor	Peptide receptors
P41587	VIPR2	Vasoactive intestinal polypeptide receptor 2	Peptide receptors
**Class B2: Adhesion (n = 16)**			
Q96PE1	ADGRA2	Adhesion G protein-coupled receptor A2	Adhesion receptors
O60241	ADGRB2	Adhesion G protein-coupled receptor B2	Adhesion receptors
Q14246	ADGRE1	Adhesion G protein-coupled receptor E1	Adhesion receptors
P48960	ADGRE5	Adhesion G protein-coupled receptor E5	Adhesion receptors
Q8IZF2	ADGRF5	Adhesion G protein-coupled receptor F5	Adhesion receptors
Q86Y34	ADGRG3	Adhesion G protein-coupled receptor G3	Adhesion receptors
O95490	ADGRL2	Adhesion G protein-coupled receptor L2	Adhesion receptors
Q9Y653	ADGRG1	Adhesion G-protein coupled receptor G1	Adhesion receptors
Q8IZP9	ADGRG2	Adhesion G-protein coupled receptor G2	Adhesion receptors
Q8IZF6	ADGRG4	Adhesion G-protein coupled receptor G4	Adhesion receptors
Q86SQ4	ADGRG6	Adhesion G-protein coupled receptor G6	Adhesion receptors
Q8WXG9	ADGRV1	Adhesion G-protein coupled receptor V1	Adhesion receptors
P08913	ADRA2A	Alpha-2A adrenergic receptor	Aminergic receptors
Q9NYQ6	CELSR1	Cadherin EGF LAG seven-pass G-type receptor 1	Adhesion receptors
Q9HCU4	CELSR2	Cadherin EGF LAG seven-pass G-type receptor 2	Adhesion receptors
Q9NYQ7	CELSR3	Cadherin EGF LAG seven-pass G-type receptor 3	Adhesion receptors
**Class C: Glutamate (n = 10)**			
Q9UBS5	GABBR1	Gamma-aminobutyric acid type B receptor subunit 1	Amino acid receptors
Q13255	GRM1	Metabotropic glutamate receptor 1	Amino acid receptors
Q14416	GRM2	Metabotropic glutamate receptor 2	Amino acid receptors
Q14832	GRM3	Metabotropic glutamate receptor 3	Amino acid receptors
Q14833	GRM4	Metabotropic glutamate receptor 4	Amino acid receptors
O15303	GRM6	Metabotropic glutamate receptor 6	Amino acid receptors
O00222	GRM8	Metabotropic glutamate receptor 8	Amino acid receptors
Q8TE23	TAS1R2	Taste receptor type 1 member 2	Sensory receptors
P59544	TAS2R50	Taste receptor type 2 member 50	Sensory receptors
Q9NYW3	TAS2R7	Taste receptor type 2 member 7	Sensory receptors
**Class F: Frizzled (n = 1)**			
Q9NPG1	FZD3	Frizzled-3	Protein receptors

**Table 2 molecules-27-06503-t002:** GPCRs with selective expression in male reproductive tissues or associated with male infertility defects. For each GPCR, selective expression in male reproductive tissues and the male infertility phenotypes observed in knock-out mice are indicated.

Gene Name	Protein Name	Tissue Expression	Male Fertility Phenotype
ADGRG1	Adhesion G-protein coupled receptor G1	Other tissues	Abnormal testis morphology; reduced male infertility
ADGRG2	Adhesion G-protein coupled receptor G2	Epididymis (enriched)	Abnormal vas deferens; azoospermia
ADGRV1	Adhesion G-protein coupled receptor V1	Other tissues	Abnormal testis morphology
ADRA2A	Alpha-2A adrenergic receptor	Other tissues	Abnormal testis morphology
CELSR1	Cadherin EGF LAG seven-pass G-type receptor 1	Seminal vesicle (enriched)	Abnormal testis morphology; reduced fertility
CYSLTR2	Cysteinyl leukotriene receptor 2	Seminal vesicle (enhanced)	N/A
EDNRA	Endothelin-1 receptor	Seminal vesicle (enhanced)	N/A
F2R	Proteinase-activated receptor 1	Other tissues	Abnormal fertility/fecundity
FZD3	Frizzled-3	Seminal vesicle (enhanced)	N/A
GRM1	Metabotropic glutamate receptor 1	Other tissues	Abnormal fertility/fecundity; male infertility; reduced fertility
GRM2	Metabotropic glutamate receptor 2	Testis (enhanced)	N/A
HTR2B	5-hydroxytryptamine receptor 2B	Other tissues	Reduced fertility
NPFFR2	Neuropeptide FF receptor 2	Seminal vesicle (enhanced)	N/A
VIPR2	Vasoactive intestinal polypeptide receptor 2	Other tissues	Abnormal testis morphology; male infertility

## Data Availability

The data presented in this study are available in Appendix A here.

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
