# Peer review of "G-Protein Coupled Receptors in Human Sperm: An In Silico Approach to Identify Potential Modulatory Targets"

_molecules, 2022, doi:10.3390/molecules27196503_

Round 1

Author Response

The manuscript by Corda et al. describes the use of computational methods to identify G protein-coupled receptors (GPCRs) and their modulatory targets in human sperm. The subject matter is certainly of interest to those who study reproductive biology and male infertility. Given the central role of GPCRs in cellular communication and their demonstrated suitability as therapeutic targets, a thorough understanding of their functional significance in the various process of sperm formation, maturation, and fertilization is warranted. The in-silico approach identified 71 GPCRs in human sperm, and amongst which some have been reported to affect male fertility. Undoubtedly, the information acquired are useful and may represent a guide for other researchers in the field. However, one important aspect that was not addressed is whether the newly identified GPCRs are actually found in human sperm. The authors need to provide results (at either mRNA or protein level) to validate the presence of at least one or two GPCRs not previously known to be expressed in human sperm. Otherwise the uncertainty of the predictive power would become a major concern. Other suggestions are given below.

Reply: We thank the reviewer for the valuable comments that we believe have contributed to improving the manuscript. Concerning the aspect highlighted by the reviewer regarding the presence of the identified GPCRs in human sperm, none of the GPCRs described were identified for the first time in this study. In fact, all the sperm GPCRs here addressed (71 GPCRs) were previously identified in proteomic studies using ejaculated human spermatozoa. Thus, we consider that given the sensibility of the technique used for their identification (mass spectrometry), their validation is not fundamental for the aim of this communication.

Major Comments

(1) In the abstract, the statement that GPCRs “regulate several physiological and pathological processes and are the target of several drugs” is very much an understatement, and should be corrected. GPCRs are involved in numerous physiological systems, and they represent the largest family of drug targets to date.

Reply: The understatement was corrected in the abstract section (lines 8-9) as suggested by the reviewer.

(2) On page 2, the authors described Gs/o as a class of G proteins. This is incorrect because Go belongs to the Gi subfamily. If they are referring to the G proteins of the olfactory system, then it should be Golf. Proper nomenclature would require the use of subscripts.

Reply: Thank you for your correction. The G proteins nomenclature was revised (line 53).

(3) Full list of the 71 GPCRs identified in human sperm is only found in Table S2. One should consider this being a major part of the results and thus should be included in the main text. The information should be organized into GPCR classes and indicate which ones are newly identified in human sperm.

Reply: We include the full list of GPCRs (former table S2) in the main results sections (now Table 1) as suggested by the reviewer. The information was organized into GPCR classes. Since all 71 GPCRs were previously identified in human sperm proteome, we consider that none of them are newly identified in human sperm.

(4) The discussion tends to focus on the comparing the list of GPCRs against the literature. This is of course needed, but it would be far more interesting to look at novel ones. What are they really expressed in human sperm? Validation by RT-PCR and Western blot analyses are thus needed.

Reply: The identified GPCRs in the performed analysis were previously identified in spermatozoa through mass spectrometry, a high sensitivity technique, confirming that these GPCRs are present in human spermatozoa. Of the 71 GPRCs identified, only four were previously studied in human spermatozoa, whereas the remaining 67 GPCRs are poorly explored in these cells. Complementary experimental validation (such as western blot, immunocytochemistry, and co-immunoprecipitation) may be useful to understand their cellular localization, role, and downstream signaling pathways in a future study. In the submitted study, the primary objective was to, based on the literature data, compile all GPCRs identified in human spermatozoa and highlight the most relevant ones for further investigation.

(5) The descriptions on interactomes are somewhat superficial. They have identified top 10 GPCRs with a higher node degree. Are these GPCRs known to play a role in human sperm biology? Examples on the involvement of a few of these GPCRs would strengthen the discussion. Also some discussion on the potential role of olfactory receptors are desirable.

Reply: The primary goal of the top 10 GPCRs was to show that sperm GPCRs may be involved in different pathways. Among the top 10, only AGTR1 was associated with capacitation, acrosome reaction, and motility in mammalian sperm. In fact, those processes are biologically distinct, which also corroborates the primary goal. As suggested by the reviewer, we added this point to the discussion (lines 191-194). Regarding olfactory receptors' roles in sperm functions, our analysis did not reveal the involvement of any of them in sperm functions.

Reviewer 2 Report

     This is a manuscript based completely on bioinformatic data mining. A proteomic database was searched for known GPCRs found in sperm. Relevant candidates were selected based on published gender-specific expression levels and genetic phenotypes. Finally interaction networks involving the identified GPCRs were identified and discussed. Thus, the manuscript does not represent any major scientific advance beyond what one might be able to accomplish with a modest amount of database searching. Nonetheless, the results of this searching do not currently appear to be available in the literature, so there may be some value in making these results available.

     I do not understand why this manuscript should be published in “Molecules” which, as far as I can tell, deals primarily with synthetic and medicinal chemistry, rather than a journal focusing more on bioinformatics or GPCRs or signaling.  

     I have to following concerns about the manuscript as currently constituted:

1) The 5% false discovery rate allowed in the proteomic searching is higher than often used in such analyses and implies that 3-4 of the presumed 71 GPCRs thought to be found in sperm may be spurious. The manuscript should contain a table with more complete proteomic identification information for the supposed sperm GPCRs, such as number of ID’d peptides and ID scores.

2) The paper jumps between databases that are sperm-specific (such as the proteomics) and databases that are male specific (expression, phenotypes, interactions). It is not at all clear that GPCRs with male-specific expression or genes with male-specific phenotypes are actually found in sperm.

3) It is troubling that the GO analysis did not uncover any sperm-specific processes, calling into question the validity of the database searches.

4) It is not clear that the protein interaction database being searched has any relevance to sperm or may reflects protein interactions seen in widely differing tissues.

5) The manuscript fails to cite or distinguish itself from a previous review on epididymal GPCRs (PMID: 32901914).

6) The manuscript contains several rhetorical and grammatical errors that should be corrected.

Author Response

This is a manuscript based completely on bioinformatic data mining. A proteomic database was searched for known GPCRs found in sperm. Relevant candidates were selected based on published gender-specific expression levels and genetic phenotypes. Finally interaction networks involving the identified GPCRs were identified and discussed. Thus, the manuscript does not represent any major scientific advance beyond what one might be able to accomplish with a modest amount of database searching. Nonetheless, the results of this searching do not currently appear to be available in the literature, so there may be some value in making these results available.

I do not understand why this manuscript should be published in “Molecules” which, as far as I can tell, deals primarily with synthetic and medicinal chemistry, rather than a journal focusing more on bioinformatics or GPCRs or signaling.  

Reply: We appreciate the reviewer’s comments which we believe will contribute to improving the manuscript. Considering the theme of the special issue “Molecular Modeling, Synthesis, and Functional Characterization of GPCR (G-Protein Coupled Receptor) Ligands" this article brings new information useful to further and better understand the GPCR/ligand involvement in male infertility using an in-silico methodology, we believe it may be suitable for publication in Molecules.

I have to following concerns about the manuscript as currently constituted:

1) The 5% false discovery rate allowed in the proteomic searching is higher than often used in such analyses and implies that 3-4 of the presumed 71 GPCRs thought to be found in sperm may be spurious. The manuscript should contain a table with more complete proteomic identification information for the supposed sperm GPCRs, such as number of ID’d peptides and ID scores.

Reply: The 5% false discovery rate (FDR) was established as the max cut-off established for the inclusion of the proteomic analyses in the study conducted by Santiago et al. (2019). Although nowadays the most prevalent FDR value is 1%, early proteomic studies fixed this value in 5%. As the sperm-proteome used resulted from a merge of more than one proteomic analysis, the same GPCR contains different ID peptides and ID scores depending on the study. Hence, to include this information in the table presented, we must include the ID peptides and ID scores from all the studies in which each GPCR was identified. Nevertheless, all the proteins included in the sperm proteome are proteins with at least 2 peptides identified in each individual study.

2) The paper jumps between databases that are sperm-specific (such as the proteomics) and databases that are male specific (expression, phenotypes, interactions). It is not at all clear that GPCRs with male-specific expression or genes with male-specific phenotypes are actually found in sperm.

Reply: Since mass spectrometry is a highly sensitive technique, it is highly likely that the identified 71 GPCRs are effectively expressed in sperm. Of these, it is important to understand which are the most relevant and potential study targets. In this sense, the other databases (such as HPA, MGI, and IMex) are relevant to establishing the potential role of these GPCRs in sperm. One of the concerns in the design of contraceptives is whether they can permanently and negatively disrupt processes involved in spermatogenesis and sperm maturation. The HPA was very useful to understand if any of the identified GPCRs is specific or enhanced in male tissues. Also, the MGI information allows us to understand whether mutations/knock out of certain GPCRs have a direct impact on male tissues and fertility. Taken together, the information retrieved from these databases allowed us to identify GPCRs for further targeting. To our best knowledge, none of these 71 GPCRs was identified as sperm-specific, neither in the databases nor in the literature.

3) It is troubling that the GO analysis did not uncover any sperm-specific processes, calling into question the validity of the database searches.

Reply: A major limitation of the bioinformatic analyses performed in spermatozoa is the lack of sperm-specific information in standard databases (such as UniProt), as well as sperm-specific databases. Hence, it was not surprising the absence of sperm-specific processes in the performed GO analysis since ClueGO uses those standard sources. Moreover, the ClueGO selection criteria define that “at least 3 genes/proteins from the uploaded list are associated with a term, and that these genes represent at least 4% of the total number of associated genes”, which also contributes to the reduction of the probability to find sperm-specific processes. As stated in the results section, only ADRA2A, AGTR1, AGTR2, FZD3 e GLP1R were implicated in sperm-related processes, but a quick search in UniProt databases showed that these GPCRs are not annotated to those sperm processes (search performed on 2nd September 2022). Being aware of these limitations, we selected the most relevant processes identified in the enrichment analysis, which may occur in sperm, mediated by the sperm GPCRs (figure 1c). We hope we have clarified this limitation in the conclusion section (lines 294-299).

4) It is not clear that the protein interaction database being searched has any relevance to sperm or may reflects protein interactions seen in widely differing tissues.

Reply: There is a lack of sperm-specific databases, including for protein-protein interactions. Through IMEx database, it was possible to retrieve the interactors of the 71 GPCRs identified in human sperm (data not shown). Cross-comparison of those interactors with the sperm proteome allowed to obtain a sub-PPI network filtered only for human spermatozoa (figure 2). Using the GO information for the sperm-interactors it was possible to infer which GPCRs may have a potential role in sperm-related processes. In fact, of the 71 GPCRs identified, only 22 were associated with sperm interactors and processes (figure 3). Indeed, further experimental confirmation of the interactions in sperm is required as well as a clarification of the downstream signaling pathways. We added this point as a limitation in the conclusion.

5) The manuscript fails to cite or distinguish itself from a previous review on epididymal GPCRs (PMID: 32901914).

Reply: The review article suggested is very interesting and, it focuses on the functions of GPCRs in the epididymis and how they have been exploited as targets. The epididymis is an essential organ for sperm maturation and has been pointed out as a potential tissue for the development of male contraception strategies. Although the review article mentions some GPCRs that were identified also by us (for example ADGRG2), it does not explore their presence and functions in sperm physiology. As stated in the introduction, our analysis was focused on the identification of GPCRs in human spermatozoa as well as their potential functions in this cell-type. Therefore, our work substantially differs from the suggested review. Nevertheless, we thank the reviewer for the suggestion, and we have included the article in the references list.

6) The manuscript contains several rhetorical and grammatical errors that should be corrected.

Reply: An exhaustive review of the English was carried out.

Reviewer 3 Report

This manuscript by Pedro O. Corda and colleagues employed an in-silico analysis to understand the G protein-coupled receptors (GPCRs) molecule in sperm functions. This reviewer does not have too many technical concerns, one remaining issue is that the authors have to discuss the bioinformatic analysis with other practical/experimental results to demonstrate the reliability by their analytical methods.

Author Response

We thank the reviewer for the comment. We agree that the bioinformatic results should be discussed with experimental results to fully understand the role of the identified GPCRs. However, this study is a brief communication that pretends to compile all GPCRs found in human sperm and highlight their relevance in sperm function to be the starting point for further investigations. 

Reviewer 4 Report

I have reviewed this paper thorughly with great interest. The authors identified 12 GPCRs in human spermatozoa and, among them, 7 had selective expression in male tissues (epididymis, 13 seminal vesicles, and testis), and 9 were associated with male infertility defects in mice.

This is an important finding and has great potential in future GPCR targeted drug design.  Further, I believe that this is an important addition to the literture. I recommend this research for publication in molecules.

One minor comment is

The limitation of this particular computational study should be added in the abstract section.

Author Response

We thank the reviewer for the comment and the minor suggestion. Due to word count, it was not possible to add the suggestion to the abstract section, but it was added in the conclusion section. We used track changes to highlight the changes made to the manuscript. Thank you for considering our manuscript for publication. 

Round 2

Reviewer 2 Report

     Although the authors have extensively revised this manuscript, it still suffers from significant deficiencies. (Numbering of the below comments is referenced to points from the previous review and author responses.) (Also, note that the .pdf of the revised manuscript showing all edits in different colors is not easy to read.)

     Relevance for this journal: Although this reviewer was previously unaware that the manuscript was being submitted for the volume entitled “Molecular Modeling, Synthesis, and Functional Characterization of GPCR (G-Protein Coupled Receptor) Ligands,” the relevance of the manuscript for this journal is still not clear, since it doesn’t focus on ligands at all and does not do “modeling, synthesis, or characterization” of ligands. Of course, relevance should be decided by the journal editors.

1) As acknowledged by the authors, their study is still based on a proteomic data set with an unusually high false discovery rate. I also understand that it would be difficult to include the relevant statistical information for each identified GPCR. However, these problems remain as serious drawbacks that prevent rigorous evaluation of the results of this study. Having two peptides identified (with unknown reliabilities) is also not a particularly strong quality control criterion. Even more concerning, in the revised version of the manuscript, the authors appear to have eliminated mention of the high false discovery rate, a serious alteration that would prevent readers from even being aware of these problems.

2) The information that 7 of the identified GPCRs had selective expression in male tissues, and 9 were associated with male infertility defects in mice is apparently not meant as confirmation that the relevant receptors are present is sperm. Thus, it does not add anything to the body of knowledge in this field, since the male specificities are already established and available in the databases.

3) The authors acknowledge that the paucity of GO terms relevant to sperm processes in the identified receptors is due, in part, to the lack of sperm-specific processes included in the GO database. Thus, the GO analysis seems to consist of linking GPCRs identified from the proteomic datasets to GO terms that the authors speculate might be involved in sperm development of function- a rather weak set of correlations.

4) The authors seem to feel that there is value in making a correlation between the GO analysis and protein interaction databases for the identified receptors, although neither of these databases is specific to sperm. Thus, I strongly agree with the their assertion that, “further experimental confirmation of the interactions in sperm is required as well as a clarification of the downstream signaling pathways.”

After re-examining the manuscript, I do not see the value of including the 825-entry Table S1 in the supplementary material, since it is a list of GPCRs that should apparently be readily available from existing databases.

Author Response

We appreciate all the Reviewer’s constructive and relevant comments during the revision process, which we believe have contributed to the improvement of the manuscript and its value. We carefully considered and answered all the questions raised and made the corrections to the manuscript when we considered them pertinent. Please find our point-by-point response below.

1) As acknowledged by the authors, their study is still based on a proteomic data set with an unusually high false discovery rate. I also understand that it would be difficult to include the relevant statistical information for each identified GPCR. However, these problems remain as serious drawbacks that prevent rigorous evaluation of the results of this study. Having two peptides identified (with unknown reliabilities) is also not a particularly strong quality control criterion. Even more concerning, in the revised version of the manuscript, the authors appear to have eliminated mention of the high false discovery rate, a serious alteration that would prevent readers from even being aware of these problems.

Reply: Thank you once again for your comment. The 5% FDR and the 2 peptides criteria were established as the inclusion criteria for the proteomic analyses in the study conducted by Santiago et al. (2019), which is, to the best of our knowledge, the most recent collection of human sperm proteome. Although nowadays the most prevalent FDR value is 1%, an updated proteome with these new criteria was not available and should be compiled in a future study. In the re-submitted version of the manuscript, we choose to maintain the information initially present “Considering that the human sperm proteome used [41] only included proteomic studies performed using ejaculated human spermatozoa in which a false discovery rate (FDR) < 5% of protein identification was set and proteins identified with at least two peptides were included, some GPCRs identified by other techniques were excluded.” (lines 80-83).

2) The information that 7 of the identified GPCRs had selective expression in male tissues, and 9 were associated with male infertility defects in mice is apparently not meant as confirmation that the relevant receptors are present is sperm. Thus, it does not add anything to the body of knowledge in this field, since the male specificities are already established and available in the databases.

Reply: One of the concerns in the design of contraceptives is whether they can permanently and negatively disrupt processes involved in spermatogenesis and sperm maturation. The HPA and MGI databases were very useful to understand if any of the identified sperm GPCRs had selective expression in male tissues or if they were previously associated with male infertility phenotypes. Thus, it was possible to highlight some sperm GPCRs (ADGRG1, ADGRV1, ADRA2A, and VIPR2) that appear to be unattractive targets since they are selective expressed in male tissues and were associated with morphological defects in KO models. Consequently, their modulation in sperm may have undesired side effects on male infertility and should be carefully evaluated in the testicular environment. These points are mentioned in the results (lines 119-123) and conclusion (lines 278-282) sections.  

3) The authors acknowledge that the paucity of GO terms relevant to sperm processes in the identified receptors is due, in part, to the lack of sperm-specific processes included in the GO database. Thus, the GO analysis seems to consist of linking GPCRs identified from the proteomic datasets to GO terms that the authors speculate might be involved in sperm development of function- a rather weak set of correlations.

Reply: A major limitation of the bioinformatic analyses performed in spermatozoa is the lack of sperm-specific information in standard databases (such as UniProt), as well as sperm-specific databases. This means that most proteins are exhaustively explored in somatic cells but are poorly characterized in spermatozoa, a highly differentiated cell with specific features. Therefore, we only can infer a putative role of the identified GPCRs in spermatozoa based on the information available in the searched databases. This limitation was mentioned in the conclusion section (lines 283-288).

4) The authors seem to feel that there is value in making a correlation between the GO analysis and protein interaction databases for the identified receptors, although neither of these databases is specific to sperm. Thus, I strongly agree with their assertion that, “further experimental confirmation of the interactions in sperm is required as well as a clarification of the downstream signaling pathways.”

Reply: We thank the reviewer for the comment. As we stated we consider that is extremely important the complementation and discussion of the bioinformatic results with experimental data to fully understand the role of the identified GPCRs.

After re-examining the manuscript, I do not see the value of including the 825-entry Table S1 in the supplementary material, since it is a list of GPCRs that should apparently be readily available from existing databases.

Reply: Although the information is readily available in those databases, we believe that Table S1 should be useful since it is a compilation of the information available in the GLASS and GPCRdb databases until March 2022. It allows a more reliable replication of the results presented or can be used to perform complementary analyses. Considering that, we decide to maintain Table S1.